# Novel Sonoguided Digital Palpation and Ultrasound-Guided Hydrodissection of the Long Thoracic Nerve for Managing Serratus Anterior Muscle Pain Syndrome: A Case Report with Technical Details

**DOI:** 10.3390/diagnostics15151891

**Published:** 2025-07-28

**Authors:** Nunung Nugroho, King Hei Stanley Lam, Theodore Tandiono, Teinny Suryadi, Anwar Suhaimi, Wahida Ratnawati, Daniel Chiung-Jui Su, Yonghyun Yoon, Kenneth Dean Reeves

**Affiliations:** 1Department of Physical Medicine and Rehabilitation, Primasatya Husada Citra Hospital, Surabaya 60165, Indonesia; nugroho.rsphc1212@gmail.com; 2Department of Physical Medicine and Rehabilitation, Faculty of Medicine, Widya Mandala Catholic University, Surabaya 60112, Indonesia; 3The Department of Clinical Research, The International Association of Musculoskeletal Medicine, Hong Kong, China; painfreedoc22@gmail.com (T.S.); buraida1303@gmail.com (W.R.); dr.daniel@gmail.com (D.C.-J.S.); mgyyh00@gmail.com (Y.Y.); 4Faculty of Medicine, The Chinese University of Hong Kong, Hong Kong, China; 5Faculty of Medicine, The University of Hong Kong, Hong Kong, China; 6The Department of Clinical Research, The Hong Kong Institute of Musculoskeletal Medicine, Hong Kong, China; 7Center for Regional Anesthesia and Pain Medicine, Wan Fang Hospital, Taipei Medical University, Taipei 110, Taiwan; 8Center for Regional Anesthesia and Pain Medicine, Chung Shan Medical University Hospital, Taichung 402, Taiwan; 9International Association of Regenerative Medicine, Namdong-gu, Incheon 21574, Republic of Korea; 10General Practitioner, Surabaya Orthopedic and Traumatology Hospital, Surabaya 60213, Indonesia; theotandiono@gmail.com; 11Department of Physical Medicine and Rehabilitation, Hermina Podomoro Hospital, North Jakarta 14350, Indonesia; 12Department of Physical Medicine and Rehabilitation, Medistra Hospital, South Jakarta 12950, Indonesia; 13Physical Medicine and Rehabilitation, Synergy Clinic, West Jakarta 11510, Indonesia; 14Department of Rehabilitation Medicine, Universiti Malaya, Kuala Lumpur 50603, Malaysia; 15Department of Neurology, Makassar Ministry of Health Central General Hospital, Makassar, Bone, South Sulawesi 90155, Indonesia; 16Department of Neurology, Tenriawaru General Hospital, Bone, South Sulawesi 90713, Indonesia; 17Department of Neurology, La Mappapenning General Hospital, Bone, South Sulawesi 90713, Indonesia; 18Department of Physical Medicine and Rehabilitation, Chi Mei Medical Center, Tainan 710, Taiwan; 19A Tempo Regeneration Center for Musicians, Tainan 700, Taiwan; 20Department of Orthopedic Surgery, Hallym University Gangnam Sacred Heart Hospital, 1 Singil-ro, Yeongdeungpo-gu, Seoul 07441, Republic of Korea; 21IncheonTerminal Orthopedics, Inha-ro 489beon-gil, Namdong-gu, Incheon 21574, Republic of Korea; 22MSKUS, 1035 E. Vista Way #128, Vista, CA 92084, USA; 23Independent Researcher, Roeland Park, KS 66205, USA; deanreevesmd@gmail.com

**Keywords:** serratus anterior muscle pain syndrome, long thoracic nerve, myofascial trigger points, chest pain, sonoguided digital palpation, ultrasound-guided hydrodissection, serratus anterior fascia, 5% dextrose in sterile water

## Abstract

**Background and Clinical Significance:** Serratus Anterior Muscle Pain Syndrome (SAMPS) is an underdiagnosed cause of anterior chest wall pain, often attributed to myofascial trigger points of the serratus anterior muscle (SAM) or dysfunction of the Long Thoracic Nerve (LTN), leading to significant disability and affecting ipsilateral upper limb movement and quality of life. Current diagnosis relies on exclusion and physical examination, with limited treatment options beyond conservative approaches. This case report presents a novel approach to chronic SAMPS, successfully diagnosed using Sonoguided Digital Palpation (SDP) and treated with ultrasound-guided hydrodissection of the LTN using 5% dextrose in water (D5W) without local anesthetic (LA), in a patient where conventional treatments had failed. **Case Presentation:** A 72-year-old male presented with a three-year history of persistent left chest pain radiating to the upper back, exacerbated by activity and mimicking cardiac pain. His medical history included two percutaneous coronary interventions. Physical examination revealed tenderness along the anterior axillary line and a positive hyperirritable spot at the mid axillary line at the 5th rib level. SDP was used to visualize the serratus anterior fascia (SAF) and LTN, and to reproduce the patient’s concordant pain by palpating the LTN. Ultrasound-guided hydrodissection of the LTN was then performed using 20–30cc of D5W without LA to separate the nerve from the surrounding tissues, employing a “fascial unzipping” technique. The patient reported immediate pain relief post-procedure, with the pain reducing from 9/10 to 1/10 on the Numeric Rating Scale (NRS), and sustained relief and functional improvement at the 12-month follow-up. **Conclusions:** Sonoguided Digital Palpation (SDP) of the LTN can serve as a valuable diagnostic adjunct for visualizing and diagnosing SAMPS. Ultrasound-guided hydrodissection of the LTN with D5W without LA may provide a promising and safe treatment option for patients with chronic SAMPS refractory to conservative management, resulting in rapid and sustained pain relief. Further research, including controlled trials, is warranted to evaluate the long-term efficacy and generalizability of these findings and to compare D5W to other injectates.

## 1. Introduction

Serratus Anterior Muscle Pain Syndrome (SAMPS) is an infrequent cause of anterior chest wall pain [1]. This syndrome is characterized by pain overlying the fifth to seventh ribs in the mid-axillary line, with referred pain that may radiate down the ipsilateral upper extremity into the palmar aspects of the ring and little fingers, as well as the lower inner side of the shoulder blade [2]. SAMPS, often attributed to myofascial trigger points of the serratus anterior muscle (SAM) or dysfunction of the Long Thoracic Nerve (LTN), can lead to significant disability, affecting ipsilateral upper limb movement and quality of life. SAMPS on the left side of the body can mimic the pain of myocardial infarction and is frequently misdiagnosed as such. The intensity of the pain is typically mild to moderate and intermittent, often described as chronic, deep, sharp, and aching in character [2]. Patients may present with weakness in the ipsilateral shoulder and upper limb, especially during flexion and abduction.

### Anatomy of SAM and LTN

The SAM originates from the outer surfaces of the first to ninth ribs and the fascia covering the external intercostal muscles. Its fibers travel posteriorly and medially around the thoracic cage, inserting along the entire length of the medial border of the scapula. The SAM is situated superficial to the first nine ribs and the fascia of the external intercostal muscles, and deep to the scapula, latissimus dorsi, trapezius, and subscapularis muscles, as well as the fascia containing the long thoracic nerve. It is also located posterior to the pectoralis major and minor muscles. The SAM contributes to the formation of the medial wall of the axilla. Notably, its lower fibers interdigitate with the upper fibers of the external abdominal oblique muscle, establishing a biotensegrity connection to the abdominal muscles [3].

SAM has two primary biomechanical functions: (1) protraction and upward rotation of the scapulothoracic joint, facilitating arm elevation through coordinated motion at both the acromioclavicular and sternoclavicular joints, and (2) dynamic stabilization of the scapula against the thoracic wall during upper limb movements [4].

The SAM receives its innervation from the LTN and derives its vascular supply from branches of the superior thoracic, lateral thoracic, and thoracodorsal arteries. The LTN arises from the anterior rami of the fifth to seventh cervical nerves (specifically the C5, C6, and C7 roots of the brachial plexus). It descends along the lateral margin of the ribs, approximately at the mid-axillary line. Traveling along the superficial surface of the Serratus Anterior muscle, the LTN innervates each of its muscle slips. Entrapment of the LTN within the fascia of the Serratus Anterior may lead to conditions such as winged scapula or SAMPS.

Due to its involvement in punching and pushing movements, the SAM is often referred to as the “boxer’s muscle.” To test its function, the patient is asked to hold their shoulder in flexion and their elbow in extension, then push their upper limb against a wall. During this action, the muscle can be visually observed and palpated. In cases where the SAM is weakened or paralyzed from LTN impingement or entrapment, the medial border of the scapula becomes prominent on the upper back, a condition known as “winged scapula” [5].

We present herein a case of chronic SAMPS in a 72-year-old male patient with three years of refractory symptoms despite conventional therapeutic interventions. To our knowledge, this represents the first documented application of sonoguided digital palpation (SDP) for definitive diagnosis, followed by successful treatment utilizing ultrasound (US)-guided hydrodissection (HD) of the LTN with 5% dextrose in water (D5W) as the sole injectate, without adjunctive local anesthetic (LA).

## 2. Case Presentation

A 72-year-old male was referred from a cardiologist to the Rehabilitation Clinic with persistent pain in the left chest radiating to the upper back. His medical history included two Percutaneous Coronary Interventions (PCIs), the first in Japan approximately 10 years ago and the second in Singapore several years ago. The patient regularly takes heart medicine prescribed by his cardiologist. The most recent CT Coronary Angiography (dated 16 December 2024) revealed a stent in the ostial proximal left anterior descending artery, with mild disease distal to the stent, mild disease at proximal RCA, and normal origins of the coronary arteries. A recent CT Thorax without contrast showed normal heart and lungs, central trachea, normal diaphragm, and no fractures of the costae, sternum, clavicle, or thoracic vertebrae.

The pain had been persisting for approximately three years, beginning after a fall during a rafting excursion. Since that incident, the patient complained of pain in the left chest, radiating to the left back. The pain had been insidious in onset, with an intensity of 6/10 on the Numeric Rating Scale (NRS) for the past three years. However, in the last two weeks, the pain had worsened to 9/10, and the Patient Specific Function Scale (PSFS) [6,7] was 11.4/50, signifying severe functional impairment after the patient pushed a trolley.

The patient reported experiencing pain on the left side of the ribcage, in the left armpit, and the lower inner side of the ipsilateral shoulder blade, with occasional radiation down the inner side of the arm (Figure 1). He described the pain as similar to that experienced during a previous heart attack, which led him to consult a cardiologist; however, examination results were found to be normal. He reported sharp pain and tenderness upon touching or pressing the affected area. The pain worsened with movements involving forward flexion or protraction of the shoulder, while relief occurred when he extended or retracted the shoulder. Prior to this rehabilitation center referral, the patient had undergone multidisciplinary specialist consultations and received multiple courses of physiotherapy combined with pharmacotherapy (including NSAIDs, pregabalin, and amitriptyline). While transient symptomatic relief was achieved through medication and manual therapy, sustained pain resolution was not obtained.

Physical examination revealed normal vital signs, with the patient reporting severe pain intensity (9/10 on the NRS) and significant functional impairment (PSFS score of 11.4/50). The patient demonstrated observable signs of distress, including frequent palmar contact with the left hemithorax. A characteristic forward head posture with associated rounded shoulders was noted, with intermittent active thoracic extension maneuvers performed to mitigate symptoms. Musculoskeletal examination of the chest wall revealed symmetrical expansion without evidence of scarring or retractions. The patient maintained a full active range of motion in both cervical spine and shoulder girdle movements, though reproduction of symptoms was noted during forward flexion and protraction of the affected shoulder. No clinical evidence of scapular winging or scapulothoracic dyskinesia was observed. Palpation elicited precise tenderness localized to the anterior axillary line at the level of the fifth rib, with a positive hyperirritable spot reproducibly demonstrated in the mid-axillary line at the corresponding rib level (Figure 1).

### 2.1. Diagnosis of SAMPS Using Sonoguided Digital Palpation

Sonoguided Digital Palpation (SDP) was employed as a diagnostic tool to visualize and assess the suspected fascial and neural lesions associated with SAMPS in this patient (Figure 2 and Figure 3, Appendix A).

This technique combines real-time ultrasound imaging with targeted layer palpation, building upon established osteopathic principles of myofascial and viscera assessment [8,9,10,11]. Traditional ultrasound palpation lacks dynamic fascial mobility assessment. Using a high-frequency linear transducer (GE Logiq E10S, linear array ML4-20-D, General Electric, Boston, MA, USA), we systematically evaluated the gliding interface between the latissimus dorsi fascia (LDF) and serratus anterior fascia (SAF), and the related thoracodorsal nerve and long thoracic nerve, respectively, at the level of the 5th rib in the mid-axillary line.

The ultrasound examination revealed distinct anatomical landmarks, including the SAM, intercostal muscles, pleura, and the neurovascular bundle containing the LTN and lateral thoracic artery (LTA) (Figure 3). Of particular diagnostic significance was the identification of restricted fascial mobility between the LDF and SAF during dynamic layer palpation, suggesting pathological adhesion formation. This finding was further corroborated by comparison with the contralateral asymptomatic side, which demonstrated normal fascial gliding. To our knowledge, this is the first study to systematically integrate layer palpation and fascial mobility assessment with conventional dynamic ultrasound palpation (or scanning).

A critical component of the SDI examination involved precise differentiation between the LTN (located within the SAF) and the nearby thoracodorsal nerve (TDN), within the LDF. This distinction was achieved through two complementary methods: First, by observing differential nerve displacement during active arm flexion/abduction, which selectively engages the latissimus dorsi muscle and its associated TDN. Second, by tracing the nerve courses proximally to identify characteristic branching patterns—the LTN was seen giving branches to the SAM, while the TDN branches supplied the latissimus dorsi. Existing methods do not systematically combine dynamic maneuvers and branching anatomy to isolate the LTN.

The diagnostic accuracy of SDP was established through a multi-component assessment protocol performed under real-time ultrasound guidance. Using a systematic approach, the examiner applied controlled digital pressure for layer palpation [8,9,10,11] at the midpoint of the transducer to selectively engage the LTN within its fascial compartment, as well as the relative movement of the LDF and SAF. This assessment incorporated conventional dynamic compression to evaluate neural tenderness, along with dynamic gliding maneuvers of layer palpation—measuring relative transverse excursions (typically approximately 5 mm)—by comparing the painful and nonpainful regions of the nerve, as well as the contralateral counterparts. This approach documented the LTN’s mobility relative to adjacent anatomical structures.

The SDP is performed using a systematic protocol requiring concordance of four predefined criteria under real-time ultrasound visualization: (1) quantitative measurement of fascial mobility during layer palpation (≤50% transverse excursion compared to the asymptomatic contralateral side), (2) precise reproduction of the patient’s characteristic pain pattern exclusively upon long thoracic nerve (LTN) provocation, and (3) identification of fascial gliding restriction. (4) identification of swelling of the LTN and loss of fascicular pattern, and an increase in hypoechogenicity and edema as documented in either compressive or inflammatory neuropathy. This multi-component diagnostic framework minimized subjectivity by anchoring findings to both patient-reported symptoms and objective sonographic metrics. Operators adhered to a systematic protocol to minimize technique variation.

The diagnostic certainty achieved through SDP offers two clinically significant advantages. First, it enables precise localization of the pathological neural-fascial interface requiring hydrodissection intervention. Second, it establishes an objective framework for post-procedural evaluation through quantitative reassessment of both fascial mobility and pain provocation thresholds. This dual diagnostic–therapeutic utility was particularly evident in our case, where pre- and post-intervention SDP findings directly correlated with the patient’s dramatic clinical improvement (Appendix A).

While nerve conduction studies/EMG could provide valuable data, they were not performed in this case as the clinical presentation (reproducible LTN tenderness on SDP with concordant pain relief post-hydrodissection) was deemed diagnostically sufficient. This aligns with current musculoskeletal practice, where ultrasound-guided diagnostics can be as accurate as electrodiagnostics for peripheral nerve entrapments or injuries [12,13,14,15].

### 2.2. Ultrasound-Guided Hydrodissection of Long Thoracic Nerve: Methodology

After confirming long thoracic nerve (LTN) entrapment through sonoguided digital palpation and obtaining informed consent, we performed ultrasound-guided hydrodissection of the affected nerve segment. The patient was positioned in lateral decubitus with the affected arm abducted approximately 90 degrees to optimize access to the axillary region while maintaining comfort (Figure 4). We implemented standard cardiac monitoring throughout the procedure and maintained strict sterile technique, with particular attention to the patient’s anticoagulation status as per cardiology recommendations.

The procedure was performed using a high-frequency linear transducer (GE Logiq E10S, linear array ML4-20-D, General Electric, Boston, MA, USA) and a 22-gauge, 2.75-inch hypodermic needle inserted in-plane from posterior to anterior. Following local anesthetic skin infiltration with 1% lidocaine, hydrodissection was initiated using 5% dextrose in water (D5W) as the sole injectate, deliberately avoiding the addition of local anesthetics to preserve accurate pain feedback during the procedure.

Under continuous ultrasound guidance, the D5W solution was carefully injected to separate the adherent fascial planes between the latissimus dorsi fascia and serratus anterior fascia, creating a characteristic fluid halo around the LTN. A critical technical aspect involved maintaining a clear visualization of the needle tip at all times while utilizing hydraulic pressure from the injectate to gently separate tissues ahead of the needle advancement [16]. This approach minimizes the risk of accidental neurovascular injury while ensuring precise delivery of the solution to the target interface.

The “fascial unzipping” technique was systematically applied, beginning at the point of maximal tenderness previously identified during diagnostic evaluation. After achieving initial decompression at the primary entrapment site, the needle was carefully pivoted from the initial entry point to address both proximal and distal segments of the entrapped nerve. The total injectate volume occasionally reached 50 mL to ensure complete circumferential nerve release, with the exact amount titrated to individual tissue response under real-time ultrasound monitoring (Figure 5). The therapeutic endpoint was determined by three objective criteria: first, direct visualization of complete fascial separation surrounding the LTN; second, immediate reduction in the patient’s pain to 3/10 or less on the numeric rating scale; and third, restoration of pain-free scapular motion, though notably no significant scapulothoracic dyskinesia was present in this particular patient.

The patient demonstrated an excellent immediate response, with pain decreasing from 9/10 to 1/10 on the numeric rating scale immediately following the procedure. Post-procedural monitoring revealed no complications beyond transient local swelling that resolved spontaneously within 10 min. Follow-up evaluations conducted at 1 week, 1, 2, 6, and 12 months confirmed sustained therapeutic benefit, with complete resolution of symptoms, significantly improved PSFS to 41.8/50 and no recurrence of pain upon SDP examination. This durable outcome suggests that the mechanical release of fascial constraints combined with the potential neuromodulatory effects of D5W may offer superior long-term results compared to traditional injectates containing local anesthetics or corticosteroids [1,17,18], as supported by emerging evidence in the literature [19,20,21,22]. The procedural details are further illustrated in Figure 4, Figure 5 and Figure 6, with corresponding video documentation available (Figure 4, Figure 5 and Figure 6 and Appendix A).

Post-procedure, the patient’s vital signs remained stable. No bruising occurred, and injection-related swelling subsided within 10 min. No complications or adverse effects were observed during the procedure or the observation period, and the patient was subsequently discharged. At follow-up appointments at 1 month, 2 months, 6 months, and 12 months, the patient reported minimal to no pain, and SDP did not elicit any positive signs. Throughout the 12-month follow-up period, the patient maintained full adherence to his prescribed cardiovascular rehabilitation regimen, with no interruptions to his exercise program.

## 3. Discussion

SAMPS is an underrecognized cause of anterior chest wall pain, often misdiagnosed due to its overlap with cardiac or rotator cuff pathology [1,23]. The syndrome arises from myofascial dysfunction of the serratus anterior muscle (SAM) or long thoracic nerve (LTN) entrapment, leading to pain along the 5th–7th ribs and referred symptoms mimicking myocardial infarction [1,2]. Diagnosis remains clinical, relying on exclusion of other conditions and physical exam findings (e.g., tenderness at the mid-axillary line, scapular dyskinesis) [1,24]. 

SAMPS, characterized by pain and dysfunction in the serratus anterior muscle, poses a diagnostic and therapeutic challenge, often stemming from LTN involvement [25]. The LTN’s critical role in serratus anterior muscle innervation underscores its importance in scapular stability and shoulder function; its dysfunction leads to muscle weakness, pain, and restricted shoulder movement [18]. As highlighted in this case, and reinforced by Bagcier and Yurdakul (2021) [26], SAMPS is an underrecognized cause of anterior chest wall pain that is frequently misdiagnosed due to its overlap with cardiac or rotator cuff pathology [1,23]. The syndrome arises from myofascial dysfunction of the serratus anterior muscle or LTN entrapment, leading to pain along the 5th–7th ribs and referred symptoms mimicking myocardial infarction [1,2], which necessitates careful differential diagnosis, especially in patients with cardiac risk factors [1,23]. Diagnosis remains clinical, relying on exclusion of other conditions and physical exam findings (e.g., tenderness at the mid-axillary line, scapular dyskinesis) [1,24].

Given the limitations of current diagnostic methods for SAMPS, SDP provides a valuable adjunct. As demonstrated in Appendix A, SDP enables real-time visualization and palpation of the serratus anterior fascia (SAF) and LTN, facilitating the reproduction of the patient’s concordant pain and differentiation from other sources of chest and shoulder pain.

While ultrasound-guided hydrodissection is an established technique [16], unlike prior LTN interventions focused on symptomatic blockade, our protocol addresses the fascial-neural pathomechanism through three synergistic innovations: (i) SDP-guided diagnosis quantifying adhesions, (ii) D5W-mediated adhesiolysis and neuromodulation, and (iii) circumferential nerve liberation via fascial unzipping. This represents a paradigm shift from palliative care to mechanistic treatment.

This case introduces three significant innovations in the management of LTN entrapment. First, we utilized 5% dextrose in water (D5W) without local anesthetic (LA), contrasting with conventional approaches that employ LA with or without steroids (e.g., Kim et al. [17]). This modification offers dual advantages: eliminating anesthetic-induced masking of incomplete nerve release while potentially harnessing dextrose’s neurotrophic and anti-inflammatory properties [20,21]. Second, our “fascial unzipping” technique systematically addresses adhesions along the entire entrapped nerve segment (Figure 5 and Figure 6, Appendix A)—unlike focal hydrodissection methods that target single sites. By pivoting the transducer and needle, we achieved circumferential release of a substantially longer LTN segment than previously described. Third, the use of real-time pain feedback (unobscured by LA effects) provided precise endpoint determination, correlating with both immediate pain reduction (9/10 to 1/10 NRS) and 12-month sustained relief. These technical refinements may explain the superior outcomes compared to traditional steroid/LA injections, which carry risks of myotoxicity, systemic absorption, and transient effects [19,20,21,22,27] (Table 1).

In summary, the integration of dynamic assessment (SDP), optimized injectate selection (D5W without LA), and standardized release techniques collectively represents a paradigm shift in managing LTN entrapment, offering both diagnostic precision and therapeutic advantages over existing methods. These advancements are further contextualized in the new comparative Table 1, which highlights how our protocol compares to static diagnostic imaging and pharmacologically-dependent treatments. As demonstrated in analogous contexts, ultrasound-guided nerve assessment exhibits high diagnostic reliability (sensitivity 97%, specificity 94%; Toia et al., 2016) [14] and excellent inter-rater agreement (κ > 0.8; Wijntjes et al., 2021) [13]. While formal inter-rater data for SDP awaits future studies, these benchmarks support the plausibility of our standardized protocol.

The successful outcome in this case reinforces the importance of a comprehensive and multifaceted approach to managing SAMPS. While conservative approaches are first-line treatments, interventional procedures may be necessary in cases refractory to conservative management, as seen in this patient who had failed conventional treatments for three years. The use of ultrasound guidance, as emphasized by Su et al. (2024), allows for precise delivery of the solution, minimizing the risk of complications and ensuring optimal targeting of the affected nerve [28].

The safety and efficacy of higher-volume hydrodissection for long thoracic nerve (LTN) decompression warrant careful consideration of both anatomical context and technical execution. While concerns regarding fascial tension and potential neuropraxia are well-founded in confined anatomical spaces (e.g., carpal or tarsal tunnels), the axillary region’s expansive fascial architecture provides distinct advantages. The intermuscular planes between the latissimus dorsi and serratus anterior muscles exhibit inherent compliance, allowing for controlled fluid dispersion without critical pressure elevation. This anatomical characteristic, combined with real-time ultrasound monitoring, justifies our use of 20–30 mL of 5% dextrose in water (D5W) for complete circumferential nerve release—a volume consistent with emerging reports for peripheral nerve hydrodissection in similarly compliant regions [19,29,30].

Our protocol prioritized safety through three evidence-based safeguards: First, volume titration in 5 mL aliquots permitted gradual fascial expansion, with ultrasound confirmation of adequate fluid dispersion before further injection. Second, dynamic patient feedback served as an early warning system for neurological irritation; any report of paresthesia triggered immediate cessation and needle repositioning. Third, endpoint-driven delivery ensured volumes were tailored to individual anatomical responses, with procedural termination upon achieving either (1) complete perineural fluid circumscription on ultrasound, or (2) resolution of pain provocation during dynamic nerve assessment. These measures collectively mitigated risks while optimizing therapeutic effect, as evidenced by our patient’s immediate pain reduction (NRS 9 to 1) and absence of neurological sequelae.

The primary rationale for using 5% dextrose in water (D5W) without LA stems from three key factors: First, while LA provides immediate nerve blockade, our therapeutic goal was not temporary conduction interruption but rather sustained neuromodulation through mechanical decompression and anti-neurogenic inflammatory effects. Second, as demonstrated in our case, D5W achieved prompt analgesia (NRS reduction from 9 to 1 within minutes), consistent with reports by Smigel et al. [31,32] and Gastons et al. [33]. describing its rapid pain-modulating properties. Third, the therapeutic benefit of D5W in neurogenic inflammation has been suggested by multiple basic science studies of Wu et al. [34] and Cherng et al. [35], so that the anti-neurogenic inflammatory properties of D5W create a transient biochemical environment that modulates neurogenic inflammation, and promote longer-term neuromodulation.

From a diagnostic and therapeutic perspective, LA-free hydrodissection provides three critical advantages:Preserved neural feedback enables real-time procedural titration. By avoiding local anesthetic-induced blockade, clinicians maintain the ability to monitor patient-reported symptoms during needle advancement and fluid injection. This continuous feedback allows for immediate adjustment of injection parameters (e.g., volume, pressure, or needle positioning) based on pain reproduction or relief, ensuring optimal nerve decompression while minimizing iatrogenic irritation.Elimination of false-negative pain response assessment. Local anesthetics can mask incomplete nerve release by temporarily suppressing pain signals, potentially leading to premature termination of hydrodissection before full adhesiolysis is achieved. Without anesthetic interference, the persistence or resolution of symptoms during the procedure provides a more reliable indicator of therapeutic efficacy.Uncompromised post-procedural evaluation. The absence of anesthetic effects permits accurate functional assessment immediately following the intervention. Clinicians can distinguish between true therapeutic success (mechanical decompression) and transient analgesia (chemical blockade), facilitating more informed decisions about further management. This is particularly valuable when assessing early outcomes or planning staged procedures.

These advantages collectively enhance the precision of both the intervention and its clinical assessment, while maintaining patient comfort through the inherent analgesic properties of dextrose-mediated hydrodissection.

Our experience since our initial consecutive patient series of hydrodissection with D5W alone [19,20], has consisted of over 5000 D5W-only hydrodissections, and we have empirically observed excellent patient tolerance while providing superior diagnostic information compared to LA-containing solutions. Our patient reported only mild discomfort during the injection, with no procedural interruptions required.

While this case demonstrates clinically significant outcomes, several methodological limitations require consideration. The single-case design inherently limits generalizability, though the patient’s 3-year refractory course provides valuable insights into treatment-resistant SAMPS. Despite employing validated outcome measures (including PSFS), the absence of a control group precludes definitive exclusion of placebo effects, which literature estimates at 30–35% for interventional pain therapies. Both SDP and hydrodissection exhibit operator dependence, and the technical complexity necessitates systematic training protocols to ensure procedural consistency. While SDP demonstrated diagnostic utility through correlation with therapeutic outcomes, formal assessment of inter-rater reliability was not conducted in this technical report. Future studies should quantify reliability using systematic video assessments evaluated by multiple blinded raters, reporting metrics such as the intraclass correlation coefficient for continuous variables (e.g., fascial mobility) and Cohen’s κ for categorical outcomes (e.g., concordant pain provocation)

Potential biases merit specific acknowledgment. While selecting a treatment-refractory case highlights therapeutic potential for complex presentations, it may not reflect typical SAMPS phenotypes. Furthermore, clinician dual roles as both interveners and assessors risk observer bias, notwithstanding mitigation efforts through systematic tools and video documentation. These limitations are counterbalanced by objective sonographic evidence of fascial release and sustained clinical improvement at the 12-month follow-up.

To establish the clinical utility of this approach, several critical future research directions should be undertaken. First, multicenter randomized controlled trials must directly compare the efficacy of 5% dextrose hydrodissection against conventional injectates (normal saline, local anesthetics, and/or corticosteroids). Such trials should incorporate three essential methodological safeguards: standardized operator training protocols to minimize technical variability, blinded independent outcome assessment to reduce confirmation bias, and comprehensive outcome metrics that combine validated functional scales (such as DASH and SPADI) with quantitative ultrasound measurements of fascial mobility.

Additional studies should systematically examine patient-specific factors that may influence treatment response, particularly given dextrose’s proposed metabolic mechanisms. These investigations should include careful evaluation of baseline metabolic status and relevant psychological comorbidities. Finally, the development and validation of standardized reliability criteria for sonoguided digital palpation through multicenter studies will be essential for establishing this technique in routine clinical practice.

## 4. Conclusions

This case report demonstrates the potential of ultrasound-guided hydrodissection with D5W without local anesthetic as a safe and effective treatment for SAMPS, particularly in cases refractory to conservative management. The diagnosis of SAMPS can be challenging due to its symptomatic overlap with conditions such as myocardial infarction and rotator cuff injuries, as evidenced by the patient’s initial presentation.

The integration of SDP as a diagnostic adjunct proved valuable in visualizing and assessing the serratus anterior fascia (SAF) and LTN, aiding in the identification of fascial defects and enhancing the overall diagnostic process. SDP offered a dynamic assessment that helped differentiate SAMPS from other sources of chest and shoulder pain. Prospective studies with larger cohorts should validate SDP’s reliability and refine its diagnostic thresholds to facilitate clinical adoption.

Ultrasound-guided hydrodissection, coupled with SDP, offers a minimally invasive and targeted approach to managing pain associated with potential LTN entrapment, facilitating improved nerve gliding and reducing local inflammation. The use of readily accessible solutions, such as D5W, further enhances its applicability in most hospitals and clinics and avoids the potential side effects of local anesthetics and steroids.

While this case suggests promising results, the findings are limited by the single-case study design. Case series or cohort-controlled studies are warranted to evaluate the long-term efficacy of D5W hydrodissection in the treatment of SAMPS. Further studies are needed to compare the outcomes and effects of various injectates, such as saline, platelet-rich plasma (PRP), and stem cells, and to explore the potential benefits of hydrodissection across different anatomical sites. Such research could significantly advance the field of musculoskeletal pain management and provide more robust evidence-based treatment guidelines for SAMPS.

## Figures and Tables

**Figure 1 diagnostics-15-01891-f001:**
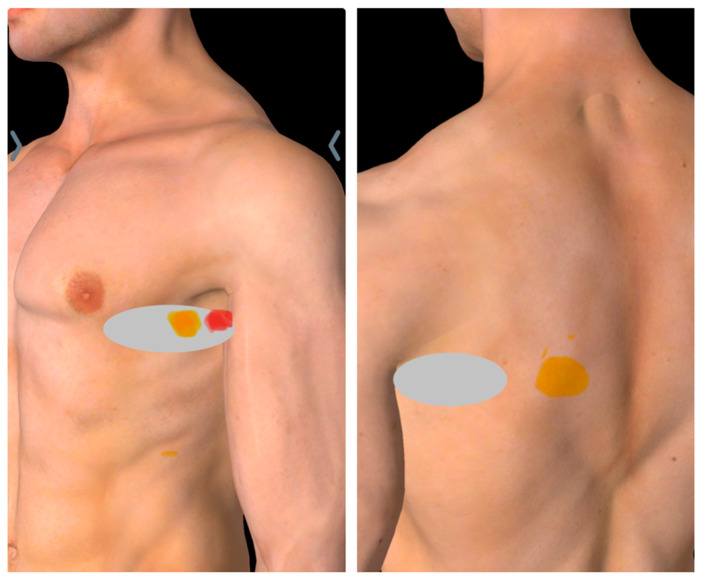
Pain mapping in serratus anterior muscle pain syndrome. Yellow markers denote tender points, red marker indicates the tender point eliciting a withdrawal response (hyperirritable spot), and gray shading demarcates the area of maximal tenderness.

**Figure 2 diagnostics-15-01891-f002:**
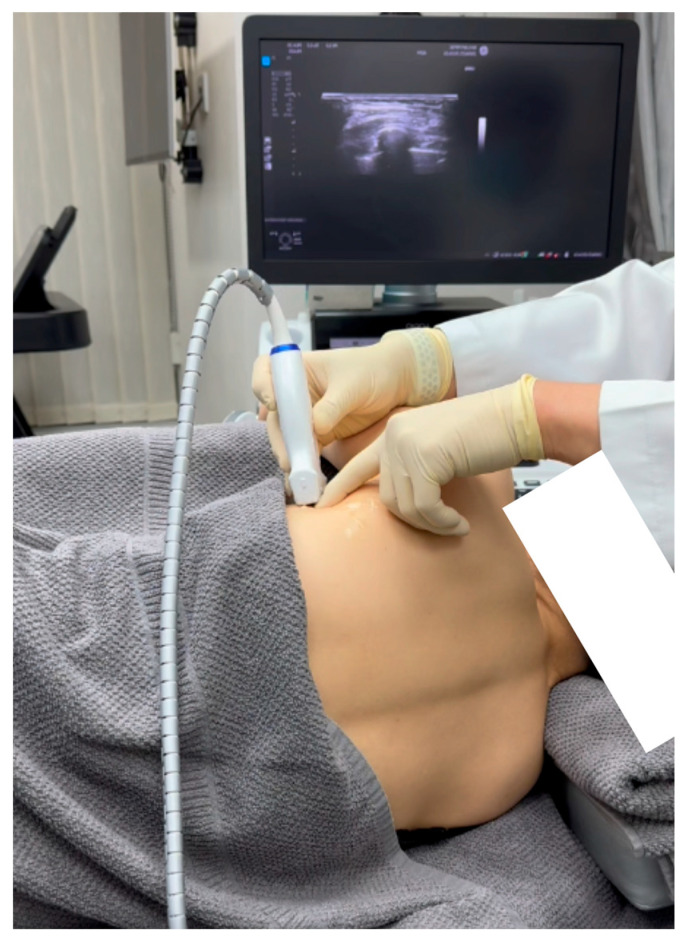
Sonoguided Digital Palpation clinical setup for serratus anterior muscle pain syndrome, illustrating optimal patient positioning and transducer placement for long thoracic nerve assessment.

**Figure 3 diagnostics-15-01891-f003:**
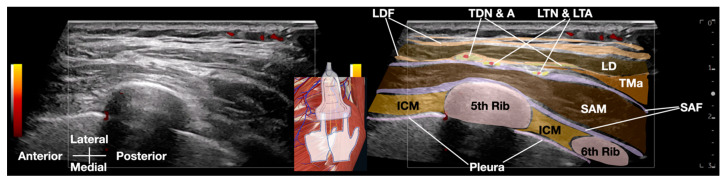
Transverse sonographic anatomy (GE Logiq E10S, ML4-20-D linear transducer, General Electric, Boston, MA, USA) at the 5th rib level for Sonoguided Digital Palpation in serratus anterior muscle pain syndrome, depicting: long thoracic nerve (LTN) within serratus anterior fascia (SAF), latissimus dorsi muscle (LD) and fascia (LDF), and key neurovascular structures. Abbreviations: ICM (intercostal muscle), LTA (lateral thoracic artery), SAM (serratus anterior muscle), TDN&A (thoracodorsal nerve and artery), TMa (teres major muscle).

**Figure 4 diagnostics-15-01891-f004:**
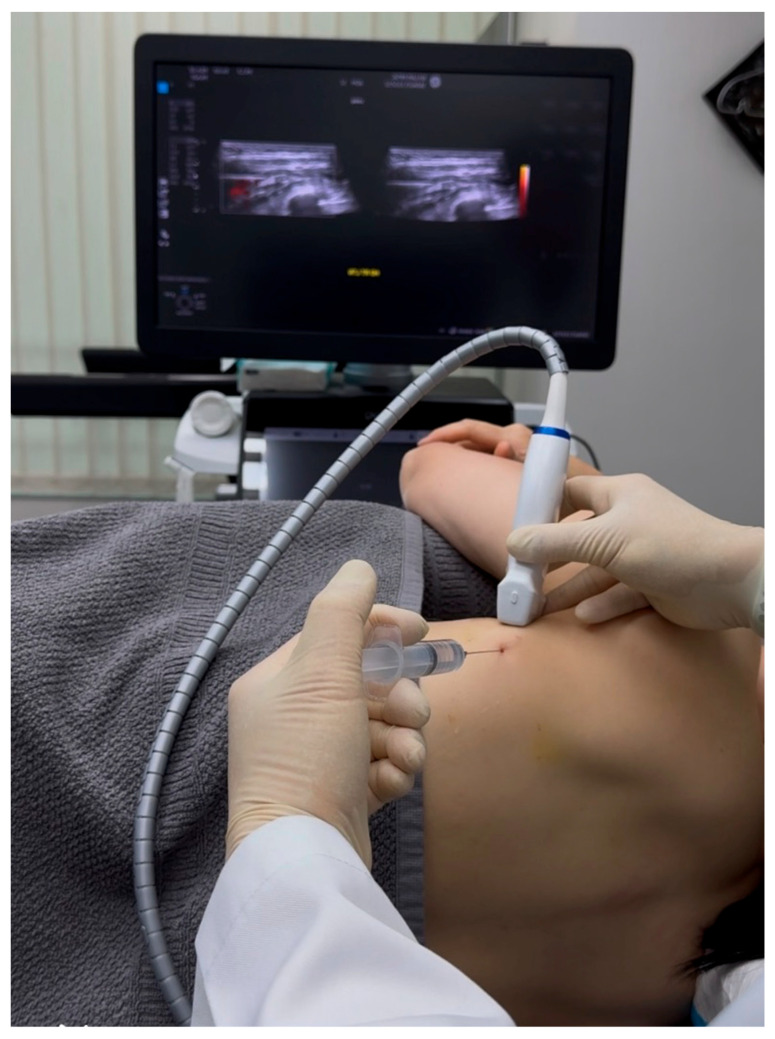
Ultrasound-guided hydrodissection setup. The patient is positioned in lateral decubitus with the affected arm abducted 90° and forward-flexed for optimal long thoracic nerve access. An in-plane posterior needle approach ensures continuous visualization of the needle trajectory.

**Figure 5 diagnostics-15-01891-f005:**
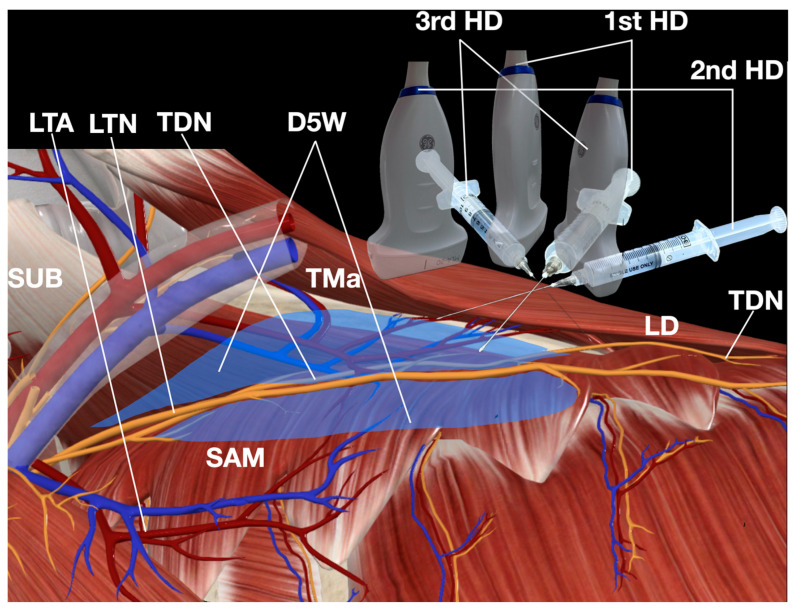
Fascial unzipping technique schematic. Abbreviations: LTN (long thoracic nerve), D5W (5% dextrose water), LD (latissimus dorsi muscle), SAM (serratus anterior muscle), LTA (lateral thoracic artery), TDN (thoracodorsal nerve), TMa (teres major muscle), SUB (subscapularis), 1st HD (first step of the hydrodissection, transducer and needle direction), 2nd HD (second step of hydrodissection, transducer and needle direction), 3rd HD (third step of hydrodissection, transducer and needle direction).

**Figure 6 diagnostics-15-01891-f006:**
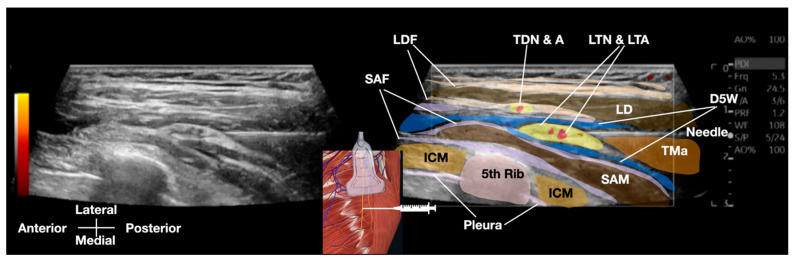
Post-hydrodissection ultrasound demonstrating successful fascial release of the long thoracic nerve (LTN) using 5% dextrose water (D5W). Visualized structures: serratus anterior muscle/fascia (SAM/SAF), latissimus dorsi muscle/fascia (LD/LDF), lateral thoracic artery (LTA), and thoracodorsal neurovascular bundle (TDN&A). Abbreviations: ICM (intercostal muscle), TMa (teres major muscle).

**Table 1 diagnostics-15-01891-t001:** Table shows the comparison of the novel diagnostic and treatment method for Serratus Anterior Muscle Pain Syndrome (SAMPS) and the conventional treatments for the SAMPS.

Feature	Bautista [1] 2017	Kim 2022 [17]	Current Study
Primary Intervention	(LA + Steroid) Trigger-point injection	Nerve block (LA + steroid)	D5W without LA hydrodissection
Diagnostic Method	Clinical palpation	Physical examination, diagnostic trigger points injection, then EMG to diagnose LTN neuropathy	Quantitative SDP
Therapeutic Target	Muscle trigger points	Anesthetic dispersion	Fascial adhesions + hydrodissecting LTN
Volume/Extent	2 mL of Bupivacaine 0.25% and Triamcinolone 4 mg/mL to each trigger points, total volume not mentioned	20 mL of bupivacaine 0.125% with 20 mg Triamcinolone in the plane between the serratus anterior and latissimus dorsi	20–30 mL D5W without LA (circumferential to the LTN and the fascial plane between the serratus anterior and latissiumus dorsi)
Mechanistic Rationale	Symptom suppression	Temporary analgesia	Biomechanical + anti-neurogenic inflammatory
Durability	Transient relief, two injections in five-month period	Need to repeat the block 7 days later, reviewed up to 3 months	12 months sustained

## Data Availability

Data related to this study has been included in the manuscript.

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
