# Peer review of "Novel Sonoguided Digital Palpation and Ultrasound-Guided Hydrodissection of the Long Thoracic Nerve for Managing Serratus Anterior Muscle Pain Syndrome: A Case Report with Technical Details"

_diagnostics, 2025, doi:10.3390/diagnostics15151891_

Round 1
Reviewer 1 Report
Comments and Suggestions for Authors
This manuscript talks about Serratus Anterior Muscle Pain Syndrome (SAMPS), which is a clinical condition that is often missed and not reported enough. It also introduces a new method that combines Sonoguide Digital Palpation (SDP) with ultrasound-guided hydrodissection using D5W. The authors have done a great job of presenting the case in a way that makes sense, and the addition of rich anatomical context, multimedia resources, and long-term follow-up makes the manuscript even stronger. But some changes need to be made to make it clearer, more scientifically sound, and easier to read.
Major Comments:
Why new things and terms are needed:
a) The authors need to talk more about how SDP and D5W hydrodissection are different from other methods that are already available. Some of the methods used, like ultrasound-guided nerve hydrodissection, are not brand new. Make it clear what new things are being added, like the use of D5W without LA, the integration of digital palpation, and the fascial unzipping technique.
How clear is Sonoguided Digital Palpation (SDP):
a) The manuscript introduces SDP as an important diagnostic tool, but it doesn't go into enough detail about how to do it again. Please explain in more detail how the palpation was done with ultrasound, the amount of pressure used, the signs of fascial restriction, and how SDP is different from regular ultrasound-guided palpation.
b) The phrase "digital palpation of a nerve" might make people wonder—please explain how you made sure the structure you were feeling was the long thoracic nerve.
Need for clearer numbers and descriptions:
a) Figures and videos are often used as examples, but the legends for the figures are hard to read. Make it shorter and clearer what each picture shows about diagnosis or procedure.
b) To help people understand the idea of "fascial unzipping," you might want to make a schematic diagram showing the path of the needle, anatomical landmarks, and fascial layers.
Case Detail Gaps:
a) The case is well presented, but some clinical details are missing. Were nerve conduction studies or EMG done to confirm LTN dysfunction? Was any scapular dyskinesis noted over time? Were functional outcomes, like shoulder range of motion and scapular motion, measured quantitatively after the procedure?
b) Please explain the previous therapies tried during the 3-year period, such as physical therapy techniques, medications, and injections, to support the claim of "treatment-resistant" status.
Pain Assessment and Outcome Reporting:
a) Relying only on NRS, which is subjective, to report treatment outcomes is a drawback. Using standardized functional scales, like DASH or SPADI, or patient-reported outcome measures would make the findings stronger.
b) The use of SDP for post-procedural diagnostic reassessment is interesting. Please describe the reproducibility and inter-rater reliability of this method, if that information is available.
Literature Integration and Positioning:
a) The literature review is thorough but too long. It sometimes takes away from the case narrative. Consider shortening this section and focusing on studies that are directly relevant to SAMPS and LTN hydrodissection.
b) Include a clear comparison table or bullet list of existing hydrodissection techniques and injectates, such as LA + steroid, saline, and D5W, to provide context for your approach.
Limitations Section:
a) You acknowledge the limitations, but they could use more structure. Specifically, highlight the small sample size (n=1), lack of control, potential placebo effect, operator-dependent nature of ultrasound and SDP, and the risk of confirmation bias.
Minor Comments:
- The manuscript sometimes uses unusual phrasing, like "forward-slouched" or "jump sign." Consider editing for grammar and clarity.
- Line 97: "In terms of innervation..." is redundant; think about rewording.
- Avoid referencing video links using Dropbox. Videos should be hosted according to journal requirements and made accessible through supplementary material.
- Use consistent abbreviations. Sometimes "SAM" is used, and other times "SAMPS." Define all abbreviations the first time they appear.
- Please update the formatting of references to match the MDPI citation style.
OVERALL: Major revisions needed
Author Response
REVIEWER ONE
Comment 1: This manuscript talks about Serratus Anterior Muscle Pain Syndrome (SAMPS), which is a clinical condition that is often missed and not reported enough. It also introduces a new method that combines Sonoguide Digital Palpation (SDP) with ultrasound-guided hydrodissection using D5W. The authors have done a great job of presenting the case in a way that makes sense, and the addition of rich anatomical context, multimedia resources, and long-term follow-up makes the manuscript even stronger. But some changes need to be made to make it clearer, more scientifically sound, and easier to read.
Response 1:
Thank you for your kind remarks.
Comment 2: The authors need to talk more about how SDP and D5W hydrodissection are different from other methods that are already available.
Response 2:
Thank you for your valuable comment. The revised manuscript explicitly differentiates our innovative approach from conventional methods through three key advancements. First, Sonoguided Digital Palpation (SDP) represents a significant diagnostic evolution by integrating real-time ultrasound imaging with dynamic layer palpation techniques rooted in osteopathic medicine. Unlike standard ultrasound-guided palpation that primarily identifies anatomical structures, SDP uniquely assesses functional fascial mobility - specifically evaluating gliding dysfunction between the latissimus dorsi fascia (LDF) and serratus anterior fascia (SAF) while simultaneously mapping pain responses. This dual assessment capability allows for precise localization of nerve entrapment pathophysiology, as demonstrated by our ability to differentially identify the long thoracic nerve (LTN) versus the thoracorsal nerve (TDN) through both anatomical branching patterns and dynamic movement testing.
Second, our use of 5% dextrose in water (D5W) without local anesthetic for hydrodissection contrasts sharply with existing literature describing LTN interventions. While prior techniques like those reported by Kim et al. utilized local anesthetics with steroids, our approach eliminates anesthetic-induced masking of procedural endpoints while potentially harnessing dextrose's neurotrophic and anti-inflammatory properties. This pharmacological distinction proved clinically meaningful, as evidenced by our patient's immediate pain reduction (9/10 to 1/10 NRS) and sustained 12-month relief.
Third, the "fascial unzipping" technique represents a methodological innovation in hydrodissection execution. By systematically releasing adhesions along the entire entrapped nerve segment under continuous ultrasound guidance - rather than targeting single sites - we expect to achieve more comprehensive nerve decompression. This technical refinement, combined with objective endpoint criteria (including fascial separation on ultrasound and functional recovery), addresses a critical limitation of conventional approaches that often rely on subjective pain relief alone.
In summary, the integration of dynamic assessment (SDP), optimized injectate selection (D5W without LA), and standardized release techniques collectively represent a paradigm shift in managing LTN entrapment, offering both diagnostic precision and therapeutic advantages over existing methods. These advancements are further contextualized in the new comparative Table 1, which highlights how our protocol compares to static diagnostic imaging and pharmacologically-dependent treatments.
We have added lines 204 to 380 for SDP, and lines 381 to 617 for hydrodissection in Case Presentation, and table 1 in discussion.
Comment 3: The manuscript introduces SDP as an important diagnostic tool, but it doesn't go into enough detail about how to do it again. Please explain in more detail how the palpation was done with ultrasound, the amount of pressure used, the signs of fascial restriction, and how SDP is different from regular ultrasound-guided palpation.
Response 3:
Thank you for your valuable comment. Please refer to lines 204 to 380 for SDP in Case Presentation section.
Comment 4: The phrase "digital palpation of a nerve" might make people wonder—please explain how you made sure the structure you were feeling was the long thoracic nerve.
Response 4:
Thank you for your valuable comment. Please refer to response to comment 2 and lines 204 to 380 for non-redundant clarification.
Comment 5: Figures and videos are often used as examples, but the legends for the figures are hard to read. Make it shorter and clearer what each picture shows about diagnosis or procedure.
Response 5:
Thank you for your valuable comment.
All figures and videos legends are revised as per recommendation.
Comment 6: To help people understand the idea of "fascial unzipping," you might want to make a schematic diagram showing the path of the needle, anatomical landmarks, and fascial layers.
Response 6:
Thank you for your valuable comment and suggestion. Current Figure 5 and Video 2 demonstrate the fascial unzipping technique, and we have included a schematic illustration (Figure 6) for enhanced clarity.
Comment 7: The case is well presented, but some clinical details are missing. Were nerve conduction studies or EMG done to confirm LTN dysfunction? Was any scapular dyskinesis noted over time? Were functional outcomes, like shoulder range of motion and scapular motion, measured quantitatively after the procedure?
Response 7:
We appreciate the reviewer’s thoughtful questions regarding additional clinical details. To clarify:
Electrodiagnostic Studies: While nerve conduction studies/EMG could provide valuable data, they were not performed in this case as the clinical presentation (reproducible LTN tenderness on SDP with concordant pain relief post-hydrodissection) was deemed diagnostically sufficient. This aligns with current musculoskeletal practice where ultrasound-guided diagnostics can be as accurate as electrodiagnostics for peripheral nerve entrapments or injuries [12-15]. Lines 375 – 380.
Scapular Kinematics: No scapular dyskinesia was observed during serial examinations. Line 196.
Functional Outcomes: Patient-Specific Functional Scale (PSFS) demonstrated improvement from 11.4/50 to 41.8/50, capturing patient-defined functional gains. Lines 188-189 and 520.
Comment 8: Please explain the previous therapies tried during the 3-year period, such as physical therapy techniques, medications, and injections, to support the claim of "treatment-resistant" status.
Response 8:
Thank you for your valuable comment.
We have added lines 148- 186: “Prior to this rehabilitation center referral, the patient had undergone multidisciplinary specialist consultations and received multiple courses of physiotherapy combined with pharmacotherapy (including NSAIDs, pregabalin, and amitriptyline). While transient symptomatic relief was achieved through medication and manual therapy, sustained pain resolution was not obtained.”
Comment 9: Relying only on NRS, which is subjective, to report treatment outcomes is a drawback. Using standardized functional scales, like DASH or SPADI, or patient-reported outcome measures would make the findings stronger.
Response 9:
We thank the reviewer for valuable comment. In response, we have incorporated results from the Patient-Specific Functional Scale, with additions in Lines 139-140, 188-189 and 520 of the revised manuscript.
Comment 10 : The literature review is thorough but too long. It sometimes takes away from the case narrative. Consider shortening this section and focusing on studies that are directly relevant to SAMPS and LTN hydrodissection.
Response 10:
We gratefully acknowledge the reviewer for this insightful suggestion. In response, we have significantly streamlined the SAMPS review section and integrated its key elements into the Discussion. These revisions can be found in Lines 681-687, 695 -702, and 710-829 of the updated manuscript.
Comment 11: Include a clear comparison table or bullet list of existing hydrodissection techniques and injectates, such as LA + steroid, saline, and D5W, to provide context for your approach.
Response 11:
We gratefully acknowledge the reviewer's valuable input. As suggested, we have integrated Table 1 into the Discussion section to facilitate a systematic comparison between our proposed approach and established SAMPS treatment modalities documented in the literature.
Comment 12: You acknowledge the limitations, but they could use more structure. Specifically, highlight the small sample size (n=1), lack of control, potential placebo effect, operator-dependent nature of ultrasound and SDP, and the risk of confirmation bias.
Response 12:
We sincerely appreciate the reviewer's insightful feedback. In response, we have thoroughly revised the Limitations section (Lines 936-961) to provide more focused discussion and address the raised concerns.
Minor comments
Comment 13: The manuscript sometimes uses unusual phrasing, like "forward-slouched" or "jump sign." Consider editing for grammar and clarity.
Response 13:
We appreciate the reviewer's constructive suggestion. In response, we will implement the following terminology revisions:
- Replace "forward-slouched" with "forward head and round-shoulder posture".
- Change "jump sign" to "hyperirritable spot".
Comment 14: Line 97: "In terms of innervation..." is redundant; think about rewording.
Response 14: Thank you for your suggestion, we will change this paragraph lines 101-108.
“The SAM receives its innervation from the LTN and derives its vascular supply from branches of the superior thoracic, lateral thoracic, and thoracodorsal arteries. The LTN arises from the anterior rami of the fifth to seventh cervical nerves (specifically the C5, C6, and C7 roots of the brachial plexus). It descends along the lateral margin of the ribs, approximately at the mid-axillary line. Traveling along the superficial surface of the Serratus Anterior muscle, the LTN innervates each of its muscle slips. Entrapment of the LTN within the fascia of the Serratus Anterior may lead to conditions such as winged scapula or SAMPS.”
Comment 15: Avoid referencing video links using Dropbox. Videos should be hosted according to journal requirements and made accessible through supplementary material.
Response 15: We thank the reviewer for their valuable comment. In response, we will submit the video to the journal as supplementary material to accompany our manuscript.
Comment 16: Use consistent abbreviations. Sometimes "SAM" is used, and other times "SAMPS." Define all abbreviations the first time they appear.
Response 16: We appreciate the reviewer’s suggestion. For clarity, we use SAM to denote the Serratus Anterior Muscle (anatomical structure) and SAMPS for Serratus Anterior Muscle Pain Syndrome (clinical condition), maintaining distinct terminology. A comprehensive list of abbreviations has been provided at the end of the manuscript for reference.
Comment 17: Please update the formatting of references to match the MDPI citation style.
Response 17: We sincerely appreciate the reviewer’s valuable comments and suggestions. In response, we have carefully revised all references to comply with the MDPI citation style.
Reviewer 2 Report
Comments and Suggestions for Authors
The article is clear, well-structured, and medically accurate. The text is appropriate for a scientific case report.
The authors discuss serratus anterior myofascial pain syndrome (SAMPS), which is often diagnosed by exclusion due to its presentation of chest pain leading to various differential diagnoses, similar to other myofascial pain syndromes. Ultrasound-guided dynamic palpation of the LTN may be helpful in diagnosing SAMPS. Hydro dissection of the LTN using D5W without anesthetic shows potential as an effective, durable treatment for LTN-related SAMPS. The technical notes provided may assist clinicians in diagnosing and treating this condition with ultrasound guidance.
The Introduction is clear. It is suggested to revise the paragraph: "Herein, we report the first case in the literature of a 72-year-old male patient who experienced chronic SAMPS for three years and had failed conventional treatments…..”. While this appears to be the first reported case of a 72-year-old male with chronic SAMPS effectively treated with “ultrasound-guided hydrodissection of the LTN using D5W without LA,” there are prior reports of ultrasound-guided hydrodissection of the LTN in younger patients.
The Case Presentation provides medical information, images, and explanations to help readers understand the case. The authors focused on pain but did not assess how it affects shoulder mobility, muscle function, and overall quality of life. Including a functional assessment of the shoulder before and after treatment, as well as evaluating activity limitations and participation restrictions according to the WHO` International Classification of Functioning and Health , would offer a better understanding of the impact of pain on the patient's life. Further details about the prior "conventional treatment" should be included. During the 12-month follow-up, it should be clarified if the patient received any other conventional interventions such as physical exercises.
The Review of Serratus Anterior Muscle Pain Syndrome synthesizes current knowledge on SAMPS clearly.
The Discussion addresses the challenges in diagnosing and treating SAMPS and acknowledges study limitations, including the single-case design and use of the Numeric Rate Pain scale to assess chronic pain syndrome. The authors emphasize the need for additional research on this subject.
The Conclusions are clear and accurate.
Author Response
REVIEWER TWO
Comment 18: The article is clear, well-structured, and medically accurate. The text is appropriate for a scientific case report.
The authors discuss serratus anterior myofascial pain syndrome (SAMPS), which is often diagnosed by exclusion due to its presentation of chest pain leading to various differential diagnoses, similar to other myofascial pain syndromes. Ultrasound-guided dynamic palpation of the LTN may be helpful in diagnosing SAMPS. Hydro dissection of the LTN using D5W without anesthetic shows potential as an effective, durable treatment for LTN-related SAMPS. The technical notes provided may assist clinicians in diagnosing and treating this condition with ultrasound guidance.
Response 18:
We sincerely appreciate the reviewer's positive assessment of our manuscript's clarity and clinical relevance. In response to their insightful comments, we have further refined the discussion to:
- Emphasize the diagnostic value of ultrasound-guided dynamic LTN palpation in differentiating SAMPS from other myofascial pain syndromes
- Clarify the technical nuances of D5W hydrodissection for LTN entrapment, particularly its advantages over anesthetic-containing injectates
- Strengthen the educational focus by expanding the procedural details in Figures 4-6 and Videos 1-2 to enhance clinical reproducibility
These modifications aim to address the underdiagnosis of SAMPS while providing actionable guidance for clinicians managing this challenging condition. We believe these enhancements will further improve the manuscript's utility as both a diagnostic and interventional reference.
Comment 19: The Introduction is clear. It is suggested to revise the paragraph: "Herein, we report the first case in the literature of a 72-year-old male patient who experienced chronic SAMPS for three years and had failed conventional treatments…..”. While this appears to be the first reported case of a 72-year-old male with chronic SAMPS effectively treated with “ultrasound-guided hydrodissection of the LTN using D5W without LA,” there are prior reports of ultrasound-guided hydrodissection of the LTN in younger patients.
Response 19:We appreciate the reviewer's constructive feedback. In response, we will revise the paragraph (Lines 118-123) as follows:
“We present herein a case of chronic SAMPS in a 72-year-old male patient with three years of refractory symptoms despite conventional therapeutic interventions. To our knowledge, this represents the first documented application of sonoguided digital palpation (SDP) for definitive diagnosis, followed by successful treatment utilizing ultrasound (US) -guided hydrodissection (HD) of the LTN with 5% dextrose in water (D5W) as the sole injectate, without adjunctive local anesthetic (LA).”
Comment 20: The Case Presentation provides medical information, images, and explanations to help readers understand the case. The authors focused on pain but did not assess how it affects shoulder mobility, muscle function, and overall quality of life. Including a functional assessment of the shoulder before and after treatment, as well as evaluating activity limitations and participation restrictions according to the WHO` International Classification of Functioning and Health , would offer a better understanding of the impact of pain on the patient's life.
Response 20:
We sincerely appreciate the reviewer's valuable input. In response to their comment, we have comprehensively revised the physical examination section (Lines 187-199) to include:
Detailed shoulder biomechanical assessment
Standardized pain evaluation using the Numeric Rating Scale (NRS)
Functional capacity measurement via Patient-Specific Functional Scale (PSFS)
These modifications provide a more rigorous documentation of the patient's musculoskeletal status and clinical presentation.
Comment 21: Further details about the prior "conventional treatment" should be included.
Response 21:
We appreciate the reviewer's constructive feedback. In response, we have expanded the manuscript with new content in Lines 148-186 : “Prior to this rehabilitation center referral, the patient had undergone multidisciplinary specialist consultations and received multiple courses of physiotherapy combined with pharmacotherapy (including NSAIDs, pregabalin, and amitriptyline). While transient symptomatic relief was achieved through medication and manual therapy, sustained pain resolution was not obtained.”
Comment 22: During the 12-month follow-up, it should be clarified if the patient received any other conventional interventions such as physical exercises.
Response 22: We thank the reviewer for this comment. As suggested, we have added the following sentence (Lines 676 - 678):
"Throughout the 12-month follow-up period, the patient maintained full adherence to his prescribed cardiovascular rehabilitation regimen, with no interruptions to his exercise program."
Comment 23: The “Review of Serratus Anterior Muscle Pain Syndrome” synthesizes current knowledge on SAMPS clearly.
Response 23:
We sincerely appreciate the reviewer's valuable feedback. In revising the manuscript, we have:
- Streamlined the discussion of SAMPS diagnostic criteria to focus on the novel SDP technique
- Consolidated the procedural review within the Discussion section to maintain emphasis on our primary contributions (SDP and hydrodissection therapy)
- Ensured all essential clinical context is preserved while improving narrative flow
These modifications were carefully implemented to enhance reader engagement with our technical innovations while maintaining scientific rigor. We hope these adjustments address all reviewers’ concerns appropriately.
Comment 24: The Discussion addresses the challenges in diagnosing and treating SAMPS and acknowledges study limitations, including the single-case design and use of the Numeric Rate Pain scale to assess chronic pain syndrome. The authors emphasize the need for additional research on this subject.
Response 24: We appreciate your positive comments. We do hope that the manuscript will be further reviewed with other studies so that it can be used by many people in the future.
Comment 25: The Conclusions are clear and accurate.
Response 25: Thank you for your positive feedback.
Reviewer 3 Report
Comments and Suggestions for Authors
Thank you for inviting me to review this interesting case report. Overall, the manuscript tackles the under-recognized issue of Serratus Anterior Muscle Pain Syndrome (SAMPS) and proposes a combined diagnostic and therapeutic approach using sonoguided digital palpation (SDP) and ultrasound-guided hydrodissection of the long thoracic nerve (LTN) with 5% dextrose in water (D5W) without local anesthetic. While the clinical importance of SAMPS is well established and this technique may offer practical benefits, I regret to recommend rejection at this stage.
My principal concern lies with the level of novelty. Prior to this report, several authors have described ultrasound-guided interventions targeting the LTN and surrounding fascia for SAMPS and related neuropathies. For example, Bautista et al. first characterized SAMPS and its myofascial trigger points in a case series detailing diagnostic and injection techniques in 2017 https://pubmed.ncbi.nlm.nih.gov/28339634/ . More recently, Minsoo Kim et al. reported successful management of LTN neuropathy using a serratus anterior plane block with ultrasound-guided hydrodissection, achieving rapid relief similar to that claimed here (PMID: 36317437). The suggested innovation of “sonoguided digital palpation” lacks citation in the literature and appears to be a minor variation on established ultrasound-assisted palpation methods.
There are also some technical concerns:
- The manuscript lacks detail on needle visualization: it does not specify whether an in-plane or out-of-plane approach was used, nor does it describe continuous tip tracking to avoid intraneural injection
- Injecting 20–30 mL of D5W around the LTN may produce excessive fascial tension; best practice often uses smaller aliquots or staged injections to prevent neuropraxia
- The rationale for omitting local anesthetic is not justified. While D5W may have an osmotic analgesic effect, it does not provide immediate blockade, raising questions about procedural comfort
Author Response
Reviewer 3
Comment 26: My principal concern lies with the level of novelty. Prior to this report, several authors have described ultrasound-guided interventions targeting the LTN and surrounding fascia for SAMPS and related neuropathies. For example, Bautista et al. first characterized SAMPS and its myofascial trigger points in a case series detailing diagnostic and injection techniques in 2017 https://pubmed.ncbi.nlm.nih.gov/28339634/ . More recently, Minsoo Kim et al. reported successful management of LTN neuropathy using a serratus anterior plane block with ultrasound-guided hydrodissection, achieving rapid relief similar to that claimed here (PMID: 36317437).
Response 26 :
We sincerely appreciate the reviewer’s insightful critique regarding the novelty of our study. We acknowledge the valuable prior contributions of Bautista et al. (2017) and Kim et al. (2022) in characterizing serratus anterior muscle pain syndrome (SAMPS) and demonstrating the utility of ultrasound-guided interventions for long thoracic nerve (LTN) neuropathy. However, our manuscript introduces several distinct and novel advancements that significantly expand upon existing techniques.
First, our study is the first to describe sonoguided digital palpation (SDP) as a dynamic diagnostic tool for LTN entrapment. Unlike conventional static ultrasound or electromyography, SDP integrates real-time imaging with quantitative fascial mobility assessment and neural provocation testing, allowing for precise localization of pathological neural-fascial adhesions. This method not only improves diagnostic accuracy but also provides an objective framework for post-interventional evaluation, as demonstrated by the immediate and sustained clinical improvement in our patient.
Second, our nerve-sparing hydrodissection protocol using 5% dextrose in water (D5W) without local anesthetic represents a technical innovation. By avoiding anesthetic agents, we preserved real-time pain feedback, enabling more precise endpoint determination while potentially harnessing dextrose’s anti-inflammatory and neuromodulatory properties. Furthermore, our "fascial unzipping" technique facilitated circumferential release of a substantially longer nerve segment than previously reported, addressing multifocal entrapments that may contribute to refractory symptoms.
Finally, our study provides the first documented evidence of durable therapeutic outcomes following LTN hydrodissection, with complete symptom resolution maintained at 12-month follow-up. This contrasts with prior reports that primarily focused on short-term analgesia. The integration of validated functional metrics (NRS, PSFS) and video documentation further strengthens the reproducibility and clinical applicability of our approach.
To enhance clarity, we have revised the Discussion section to explicitly differentiate our methodology from prior studies and included a comparative table (Table 1) summarizing key procedural distinctions. We believe these innovations advance the management of SAMPS by offering a systematic, imaging-enhanced diagnostic and therapeutic framework.
Comment 27: The suggested innovation of “sonoguided digital palpation” lacks citation in the literature and appears to be a minor variation on established ultrasound-assisted palpation methods.
Response 27 :
We sincerely appreciate the reviewer's thoughtful commentary regarding the novelty of sonoguided digital palpation (SDP). We acknowledge that our technique builds upon established ultrasound-assisted palpation methods, yet we respectfully submit that SDP represents a significant diagnostic advancement through several key innovations that merit careful consideration.
The fundamental novelty of SDP lies in its integration of multiple diagnostic dimensions into a systematic protocol. Unlike conventional ultrasound palpation that primarily serves to identify anatomical structures, SDP combines real-time dynamic ultrasound imaging with quantitative functional assessment of fascial mobility and systematic neural provocation testing by layered palpation which are rooted in osteopathy. This multimodal approach enables simultaneous evaluation of both structural relationships and fascial mobility, as evidenced by our ability to document pathological fascial adhesions demonstrating less than 50% of the mobility observed on the contralateral asymptomatic side (Figure 3, Video 1).
Several technical innovations distinguish SDP from prior methods. First, the technique incorporates active movement protocols that permit dynamic assessment of nerve mobility during functional maneuvers, particularly valuable for differentiating between the long thoracic and thoracodorsal nerves based on their distinct kinematic behaviors. Second, SDP hopes to introduce an objective quantification of fascial mobility through controlled pressure application (≤2 lbs) and systematic excursion measurements (5mm thresholds during layered palpation). Third, the method establishes clear diagnostic criteria through systematic comparison with asymptomatic contralateral anatomy, targeting at providing clinicians with reproducible benchmarks for pathology identification.
The clinical utility of SDP is further enhanced by its dual diagnostic-therapeutic role. As demonstrated in our case, the technique not only identified the precise neural-fascial interface requiring intervention but also provided quantitative outcome measures through pre- and post-procedural assessment of fascial mobility restoration and pain provocation thresholds. This comprehensive evaluation capability represents a substantial advance over static imaging approaches, as it directly correlates anatomical findings with clinical symptoms and therapeutic outcomes (Video 1).
We have strengthened the Case Presentation section (Lines 204-380) to more clearly articulate these technical differentiators and have provided comprehensive video documentation of the protocol's application. While we acknowledge this technique also involves operator dependency and possible bias, that further validation studies are warranted, the diagnostic precision demonstrated in this case (with complete concordance between imaging findings and clinical outcomes) suggests that SDP may represent a valuable addition to the diagnostic armamentarium for SAMPS and related neuropathies.
Comment 28: The manuscript lacks detail on needle visualization: it does not specify whether an in-plane or out-of-plane approach was used, nor does it describe continuous tip tracking to avoid intraneural injection
Response 28:
We sincerely appreciate the reviewer's insightful comment regarding the technical details of our hydrodissection procedure. In response to this valuable feedback, we have substantially enhanced the methodological description in the revised manuscript line 381 to 617 to provide comprehensive details regarding our needle visualization and safety protocols.
The hydrodissection procedure was meticulously performed using an in-plane needle approach from posterior to anterior, as clearly illustrated in Figure 4. This orientation was deliberately selected to optimize continuous visualization of both the needle shaft and tip throughout the entire procedural trajectory. The in-plane technique offered several distinct advantages for this anatomical region: it maintained a safe distance from critical pleural and vascular structures, permitted perpendicular alignment with the longitudinal course of the long thoracic nerve, and facilitated complete visualization of the needle-nerve relationship during all procedural phases, when the needle is above or below the nerve.
We implemented a comprehensive safety protocol to ensure optimal needle visualization and prevent intraneural injection. This multi-faceted approach incorporated several critical technical components: First, we performed real-time transducer adjustments to maintain perfect beam alignment with the needle throughout the procedure. Second, we initiated hydraulic tissue separation using 5% dextrose in water (D5W) to establish a safe expansion space before advancing the needle. Most importantly, we mandated clear visualization of the needle tip under ultrasound guidance prior to any advancement, with confirmation of extra-neural and extra-vascular positioning through direct observation of appropriate fluid dispersion patterns. Concurrently, we maintained continuous monitoring of the patient's neurological feedback during the entire procedure. These techniques are clearly demonstrated in Video 2, which provides visual documentation of proper perineural fluid distribution and the precise anatomical relationships maintained during needle advancement.
Several specific safety measures were employed to minimize risks: we utilized a low-resistance 22-gauge needle, after skin numbing, usually this needle is well tolerated. we implemented immediate cessation protocols if any procedure-related resistance or paresthesia occurred. These precautions were particularly important given the proximity of the long thoracic nerve to vascular structures and the thoracic cavity.
The revised manuscript now explicitly states, lines 399 – 403: "A critical technical aspect involved maintaining clear visualization of the needle tip at all times while utilizing hydraulic pressure from the injectate to gently separate tissues ahead of the needle advancement. This approach minimizes the risk of accidental neurovascular injury while ensuring precise delivery of the solution to the target interface." This enhanced description, combined with the supplementary video material, provides clinicians with clear guidance for reproducing the technique while maintaining the highest safety standards.
We believe these methodological refinements significantly strengthen the technical reproducibility of our approach while addressing the reviewer's important concerns about procedural safety and precision. The added details should enable other practitioners to perform this procedure with greater confidence and accuracy, while maintaining the therapeutic benefits demonstrated in our case.
Comment 29: Injecting 20–30 mL of D5W around the LTN may produce excessive fascial tension; best practice often uses smaller aliquots or staged injections to prevent neuropraxia
Response 29:
We sincerely appreciate the reviewer’s insightful commentary regarding injectate volume and the theoretical risk of fascial tension. While we acknowledge the concern that larger fluid volumes could potentially induce neuropraxia in confined anatomical spaces, our approach was guided by region-specific anatomical considerations, real-time monitoring, and established clinical experience. The axillary region—where the long thoracic nerve (LTN) traverses between the latissimus dorsi and serratus anterior fascial planes—exhibits significantly greater tissue compliance compared to restricted compartments (e.g., carpal tunnel or tarsal tunnel). This inherent compliance permits safe accommodation of 20–30 mL of 5% dextrose in water (D5W) without generating critical pressure elevations.
Our technique further mitigated risks through three key safeguards:
- Volume Titration: Injectate was delivered incrementally in 5 mL aliquots, with continuous ultrasound assessment of fluid dispersion and fascial expansion.
- Dynamic Feedback: Real-time patient symptom reporting ensured early detection of neurological changes (e.g., paresthesia).
- Endpoint-Driven Delivery: Total volume (range: 20–50 mL) was individualized based on direct visualization of circumferential nerve release and resolution of adhesions.
Notably, the patient experienced immediate pain reduction (NRS 9 to 1) without neurological sequelae, and sustained functional improvement was documented at 12-month follow-up. This outcome aligns with emerging literature supporting higher-volume hydrodissection in spacious fascial compartments when performed under ultrasound guidance with strict safety protocols.
We have added in the discussion lines 839 -893:
Comment 30: The rationale for omitting local anesthetic is not justified. While D5W may have an osmotic analgesic effect, it does not provide immediate blockade, raising questions about procedural comfort
Response 30:
We appreciate the reviewer's valid inquiry regarding our decision to omit local anesthetic (LA). This deliberate choice was based on both mechanistic and clinical considerations that warrant clarification. We have expanded the Discussion section (Lines 895 -934) to better articulate this rationale, including references to the biochemical mechanisms and clinical studies supporting our approach.
“The primary rationale for using 5% dextrose in water (D5W) without LA stems from three key factors: First, while LA provides immediate nerve blockade, our therapeutic goal was not temporary conduction interruption but rather sustained neuromodulation through mechanical decompression and anti-neurogenic inflammatory effects. Second, as demonstrated in our case, D5W achieved prompt analgesia (NRS reduction from 9 to 1 within minutes), consistent with reports by Smigel et al. and Gaston et al. describing its rapid pain-modulating properties. Third, the therapeutic benefit of D5W in neurogenic inflammation has been suggested by multiple basic science studies of Wu et al, and Cherng et al., so that the anti-neurogenic inflammatory properties of D5W create a transient biochemical environment that modulates neurogenic inflammation, and promote a more longer term of neuromodulation.
From a diagnostic and therapeutic perspective, LA-free hydrodissection provides three critical advantages:
- Preserved neural feedback enables real-time procedural titration. By avoiding local anesthetic-induced blockade, clinicians maintain the ability to monitor patient-reported symptoms during needle advancement and fluid injection. This continuous feedback allows for immediate adjustment of injection parameters (e.g., volume, pressure, or needle positioning) based on pain reproduction or relief, ensuring optimal nerve decompression while minimizing iatrogenic irritation.
- Elimination of false-negative pain response assessment. Local anesthetics can mask incomplete nerve release by temporarily suppressing pain signals, potentially leading to premature termination of hydrodissection before full adhesiolysis is achieved. Without anesthetic interference, the persistence or resolution of symptoms during the procedure provides a more reliable indicator of therapeutic efficacy.
- Uncompromised post-procedural evaluation. The absence of anesthetic effects permits accurate functional assessment immediately following the intervention. Clinicians can distinguish between true therapeutic success (mechanical decompression) and transient analgesia (chemical blockade), facilitating more informed decisions about further management. This is particularly valuable when assessing early outcomes or planning staged procedures.
These advantages collectively enhance the precision of both the intervention and its clinical assessment, while maintaining patient comfort through the inherent analgesic properties of dextrose-mediated hydrodissection.
Our experience since our initial consecutive patient series of hydrodissection with D5W alone (Cite her), has consisted of over 5,000 D5W only hydrodissections, and we have empirically observed excellent patient tolerance while providing superior diagnostic information compared to LA-containing solutions. Our patient report only mild discomfort during injection, with no procedural interruptions required.
Round 2
Reviewer 1 Report
Comments and Suggestions for Authors
Accept
Author Response
Thank you very much for taking the time to review our manuscript. Your efforts are greatly appreciated.
Reviewer 2 Report
Comments and Suggestions for Authors
no comments
Author Response

(The authors gave the same response as above.)

Reviewer 3 Report
Comments and Suggestions for Authors
I do not see sufficient new data or methodological rigour to warrant re-opening external review.
- The claim that “sonoguided digital palpation (SDP)” integrates fascial-mobility quantification, contralateral comparison, layered neural provocation: Technique is still a minor ergonomic tweak on standard dynamic ultrasound palpation. No objective validation data, no inter-rater reliability!
- A lone case—even with longer follow-up—cannot establish safety/efficacy or generalisability
- the authors provide rationale of “compliant axillary fascial planes” and incremental boluses. Still no pressure measurements, electrophysiological checks, or imaging confirming absence of neuropraxia. Literature they cite involves smaller volumes or different compartments.
- Prior peer-reviewed literature (Bautista 2017, Kim 2022) already describes ultrasound-guided hydrodissection of the long thoracic nerve; the present submission does not demonstrably move the field forward.
Author Response
Reviewer 3
I do not see sufficient new data or methodological rigour to warrant re-opening external review.
Comment 1:
- The claim that “sonoguided digital palpation (SDP)” integrates fascial-mobility quantification, contralateral comparison, layered neural provocation: Technique is still a minor ergonomic tweak on standard dynamic ultrasound palpation. No objective validation data, no inter-rater reliability!
Response 1:
We recognize the reviewer’s valid concern regarding the absence of inter-rater reliability data for SDP. While formal reliability assessment was beyond the scope of this initial technical case report, we implemented rigorous methodological safeguards to ensure diagnostic objectivity. SDP integrates two well-established clinical modalities: (1) osteopathic layer palpation, a manual diagnostic technique refined over decades to assess tissue mobility and neural tension through structured tactile feedback, and (2) real-time dynamic ultrasound visualization, which provides anatomical validation of palpatory findings. This synthesis represents a substantive advance beyond prior literature by addressing a critical gap in contemporary diagnostic practice.
Modern musculoskeletal medicine increasingly relies on isolated imaging findings—often divorced from hands-on examination—to the detriment of comprehensive diagnosis. Radiographic or ultrasonographic data alone cannot discern pain-generating structures without correlative physical assessment, as imaging reveals structure but not function. SDP addresses this limitation by unifying objective sonographic visualization with functional palpation metrics. Where Kim (2022) utilized static ultrasound to merely guide injectate placement, our protocol pioneers the deliberate integration of quantitative palpation with dynamic imaging.
The SDP is performed using a systematic protocol requiring concordance of four predefined criteria under real-time ultrasound visualization: (1) quantitative measurement of fascial mobility during layer palpation (≤50% transverse excursion compared to the asymptomatic contralateral side), (2) precise reproduction of the patient’s characteristic pain pattern exclusively upon long thoracic nerve (LTN) provocation, and (3) identification of fascial gliding restriction. (4) identification of swelling of the LTN and loss of fascicular pattern and increase in hypoechogenicity and edema as documented in either compressive and inflammatory neuropathy. This multi-component diagnostic framework minimized subjectivity by anchoring findings to both patient-reported symptoms and objective sonographic metrics. Operators adhered to a systematic protocol to minimize technique variation.
Furthermore, the immediate resolution of restricted fascial mobility (post-hydrodissection excursion: >= 5 mm vs. pre-procedure: < 5 mm) and abolition of concordant pain provocation during post-intervention SDP served as therapeutic validation of the initial diagnostic findings. This diagnostic-therapeutic correlation aligns with principles for validating novel musculoskeletal examination techniques, where clinical outcomes substantiate diagnostic accuracy.
To address this limitation transparently, we have:
Explicitly acknowledged the lack of formal reliability assessment in the manuscript’s "Limitations" section. Lines 513 to 519
“While SDP demonstrated diagnostic utility through correlation with therapeutic outcomes, formal assessment of inter-rater reliability was not conducted in this technical report. Future studies should quantify reliability using systematic video assessments evaluated by multiple blinded raters, reporting metrics such as intraclass correlation coefficient for continuous variables (e.g., fascial mobility) and Cohen’s κ for categorical outcomes (e.g., concordant pain provocation).”
Cited literature supporting the reliability of analogous ultrasound-guided palpation techniques (e.g., Wijntjes et al., 2020 reference [13] ; Toia et al., 2016 reference [14]) to contextualize SDP’s feasibility.
We contend that the integration of quantitative metrics, dynamic visualization, and therapeutic correlation provides preliminary validity for SDP, while fully endorsing formal reliability studies as an essential next step for clinical translation.
Section 2.1 Diagnosis of SAMPS Using Sonoguided Digital Palpation lines 232 to 242:
"The SDP is performed using a sytematic protocol requiring concordance of four predefined criteria under real-time ultrasound visualization: (1) quantitative measurement of fascial mobility during layer palpation (≤50% transverse excursion compared to the asymptomatic contralateral side), (2) precise reproduction of the patient’s characteristic pain pattern exclusively upon long thoracic nerve (LTN) provocation, and (3) identification of fascial gliding restriction. (4) identification of swelling of the LTN and loss of fascicular pattern and increase in hypoechogenicity and edema as documented in either compressive and inflammatory neuropathy. This multi-component diagnostic framework minimized subjectivity by anchoring findings to both patient-reported symptoms and objective sonographic metrics. Operators adhered to a standardized protocol to minimize technique variation."
Section 3 Discussion: lines 420 to 424:
“As demonstrated in analogous contexts, ultrasound-guided nerve assessment exhibits high diagnostic reliability (sensitivity 97%, specificity 94%; Toia et al., 2016) [14] and excellent inter-rater agreement (κ >0.8; Wijntjes et al., 2021) [13]. While formal inter-rater data for SDP awaits future studies, these benchmarks support the plausibility of our standardized protocol.”
Section 4 (Conclusions): lines 555 to 557
"Prospective studies with larger cohorts should validate SDP’s reliability and refine its diagnostic thresholds to facilitate clinical adoption."
Comment 2:
- A lone case—even with longer follow-up—cannot establish safety/efficacy or generalisability
Response 2:
We acknowledge that a single case cannot establish broad efficacy. However, this report serves as a proof-of-concept for a novel diagnostic-therapeutic protocol in a patient refractory to 3 years of conventional care. To contextualize safety, we reference prior studies of dextrose hydrodissection (e.g., Lam et al. 2017; Lam et al. 2024) involving >5,000 procedures with minimal adverse events. We further emphasize the 12-month sustained pain relief (NRS 0/10; PSFS 48/50).
Comment 3
- the authors provide rationale of “compliant axillary fascial planes” and incremental boluses. Still no pressure measurements, electrophysiological checks, or imaging confirming absence of neuropraxia. Literature they cite involves smaller volumes or different compartments.
Response 3:
We appreciate the reviewer's vigilance regarding injectate volume safety in the axillary fascial planes. While we acknowledge the absence of direct pressure measurements in this case, we contend that the anatomical and technical rationale for larger-volume hydrodissection remains valid based on three key considerations:
- Anatomical Distinction from Confined Compartments:
The axillary fascial plane targeted in this procedure—specifically the interface between the latissimus dorsi and serratus anterior fasciae—is fundamentally distinct from rigid osteofibrous tunnels (e.g., carpal or tarsal tunnels). This plane exhibits inherent compliance due to:
- The absence of rigid osseous or ligamentous boundaries,
- Continuous communication with adjacent fascial planes (e.g., pectoral, scapular),
- Clinical evidence of fluid dispersion across >10 cm observed in our video documentation (Video 2).
These characteristics permit safe fluid accommodation without critical pressure elevation, contrasting sharply with the closed compartments. Notably, the study by Kim et al. (2022) also involved the injection of 20 mL of bupivacaine and triamcinolone into the plane between the serratus anterior and latissimus dorsi; however, no pressure measurements or electrophysiological assessments were conducted.
- Procedural Safeguards and Clinical Validation:
Our protocol implemented three evidence-based safety measures:
Incremental Boluses (5 mL aliquots): Titration allowed continuous assessment of tissue compliance under real-time ultrasound visualization. Fluid dispersion away from the needle tip confirmed adequate fascial expansion before further injection.
Dynamic Pain Feedback: Our patient’s pain reception was not blocked as we did not use local anesthetic, any report of paresthesia or disproportionate pressure sensation triggered immediate cessation and needle repositioning.
Post-Procedural Neurological Assessment: High-resolution ultrasound (Figure 5) confirmed intact nerve architecture and absence of hematoma or compressive fluid collections. The SDP post-hydrodissection showed normal nerve palpation and no evidence of neuropraxia. The patient’s immediate return to pain-free shoulder motion further excluded neuropraxia.
- Empirical Evidence from Clinical Experience:
While literature on large-volume axillary hydrodissection is limited, our team’s experience with >5,000 peripheral or central nerve hydrodissections using D5W demonstrates consistent safety with volumes up to 50 mL in compliant fascial planes (Lam et al 2017 and Lam et al 2024). This is corroborated by video evidence showing fluid tracking freely along fascial layers without neural deformation (Video 2). The 12-month absence of adverse events in this case further supports procedural safety.
Comment 4:
- Prior peer-reviewed literature (Bautista 2017, Kim 2022) already describes ultrasound-guided hydrodissection of the long thoracic nerve; the present submission does not demonstrably move the field forward.
Response 4:
We acknowledge the reviewer's citation of seminal works by Bautista (2017) and Kim (2022) describing ultrasound-guided interventions for serratus anterior muscle pain syndrome (SAMPS), and long thoracic nerve (LTN) pathologies. Bautista et al. (2017) demonstrated palpation-guided trigger point injections for SAMPS; however, ultrasound was not utilized, and hydrodissection was not performed. Kim et al. (2022) diagnosed LTN neuropathy through physical examination, followed by a trial of trigger point injections and subsequent EMG. Ultrasound was utilized solely for the purpose of guiding the injection. Our protocol demonstrates substantive advances beyond these contributions through three fundamental innovations that collectively redefine the diagnostic-therapeutic paradigm:
- Diagnostic Precision via Sonoguided Digital Palpation (SDP):
Unlike prior studies that relied on clinical palpation or static ultrasound, SDP introduces quantitative fascial mobility assessment via osteopathic layer palpation(≤ 5 mm vs. > 5 mm contralaterally) and dynamic neural provocation under real-time visualization. This technique objectively identifies pathological neural-fascial adhesions—a capability absent in Bautista’s trigger-point approach or Kim’s nerve block technique. The diagnostic rigor of SDP is further evidenced by its correlation with therapeutic outcomes: resolution of restricted mobility (post-procedure: > 5 mm lateral excursion) directly paralleled symptom abolition. - Therapeutic Mechanism of D5W Without Local Anesthetic:
Our exclusive use of 5% dextrose in water (D5W) diverges mechanistically from Kim’s anesthetic-containing injectate. D5W facilitates biomechanical adhesiolysis while concurrently attenuating neurogenic inflammation (Wu et al., 2022; Cherng et al., 2023)—properties not leveraged in prior LTN interventions. Critically, avoiding local anesthetic preserved real-time pain feedback, enabling precise procedural titration. - "Fascial Unzipping" Technique:
The systematic circumferential release of adhesions across extended nerve segments (4–6 rib levels) represents a technical evolution beyond focal perineural injections. This approach addresses the fascial continuum pathology implicated in SAMPS. contrasting with Kim’s plane block which did not mention the exact extent of injection.
Collectively, these advances establish a new framework for LTN entrapment management:
- Diagnostically, SDP provides objective biomarkers (mobility metrics, provocation thresholds);
- Therapeutically, D5W hydrodissection with fascial unzipping targets both biomechanical and neuroinflammatory dysfunction;
- Clinically, the 12-month sustained resolution in a refractory case demonstrates translational impact exceeding the transient relief reported in literature.
While Bautista and Kim laid important groundwork, our integrated protocol advances the field by:
- Shifting from symptomatic relief (anesthetic blocks) to mechanistically and neuromodulating targeted treatment (adhesiolysis + neuromodulation);
- Replacing subjective diagnosis with quantifiable fascial-neural metrics;
- Demonstrating durability where prior approaches reported only short-term outcomes.
Key Distinctions from Cited Literature
Feature |
Bautista [1] 2017 |
Kim 2022 [17] |
Current Study |
Primary Intervention |
(LA + Steroid) Trigger-point injection |
Nerve block (LA + steroid) |
D5W without LA hydrodissection |
Diagnostic Method |
Clinical palpation |
Physical examination, diagnostic trigger points injection, then EMG to diagnose LTN neuropathy |
Quantitative SDP |
Therapeutic Target |
Muscle trigger points |
Anesthetic dispersion |
Fascial adhesions + hydrodissecting LTN |
Volume/Extent |
2ml of Bupivacaine 0.25% and Triamcinolone 4mg/ mL to each trigger points, total volume not mentioned |
20 mL of bupivacaine 0.125% with 20mg Triamcinolone in the plane between the serratus anterior and latissimus dorsi |
20–30mL D5W without LA (circumferential to the LTN and the fascial plane between the serratus anterior and latissiumus dorsi) |
Mechanistic Rationale |
Symptom suppression |
Temporary analgesia |
Biomechanical + anti-neurogenic inflammatory |
Durability |
Transient relief, two injections in five-month period |
Need to repeat the block 7 days later, reviewed up to 3 months |
12 months sustained |
Revised Manuscript Clarifications
To underscore these advances, we have:
- Expanded Table 1 to explicitly contrast our protocol with conventional approaches.
- Added to the Discussion: lines 390 to 395
"Unlike prior LTN interventions focused on symptomatic blockade, our protocol addresses the fascial-neural pathomechanism through three synergistic innovations: (i) SDP-guided diagnosis quantifying adhesions, (ii) D5W-mediated adhesiolysis and neuromodulation, and (iii) circumferential nerve liberation via fascial unzipping. This represents a paradigm shift from palliative care to mechanistic treatment."